# Automated Extraction of Annual Erosion Rates for Arctic Permafrost Coasts Using Sentinel-1, Deep Learning, and Change Vector Analysis

**Marius Philipp** [1,2,*], **Andreas Dietz** [2], **Tobias Ullmann** [3] and **Claudia Kuenzer** [1,2]

1   Department of Remote Sensing, Institute of Geography and Geology, University of Wuerzburg,
    D-97074 Wuerzburg, Germany; claudia.kuenzer@dlr.de
2   German Remote Sensing Data Center (DFD), German Aerospace Center (DLR), Muenchner Strasse 20,
    D-82234 Wessling, Germany; andreas.dietz@dlr.de
3   Department of Physical Geography, Institute of Geography and Geology, University of Wuerzburg,
    D-97074 Wuerzburg, Germany; tobias.ullmann@uni-wuerzburg.de
*   Correspondence: marius.philipp@uni-wuerzburg.de

**Abstract:** Arctic permafrost coasts become increasingly vulnerable due to environmental drivers such as the reduced sea-ice extent and duration as well as the thawing of permafrost itself. A continuous quantification of the erosion process on large to circum-Arctic scales is required to fully assess the extent and understand the consequences of eroding permafrost coastlines. This study presents a novel approach to quantify annual Arctic coastal erosion and build-up rates based on Sentinel-1 (S1) Synthetic Aperture RADAR (SAR) backscatter data, in combination with Deep Learning (DL) and Change Vector Analysis (CVA). The methodology includes the generation of a high-quality Arctic coastline product via DL, which acted as a reference for quantifying coastal erosion and build-up rates from annual median and standard deviation (sd) backscatter images via CVA. The analysis was applied on ten test sites distributed across the Arctic and covering about 1038 km of coastline. Results revealed maximum erosion rates of up to 160 m for some areas and an average erosion rate of 4.37 m across all test sites within a three-year temporal window from 2017 to 2020. The observed erosion rates within the framework of this study agree with findings published in the previous literature. The proposed methods and data can be applied on large scales and, prospectively, even for the entire Arctic. The generated products may be used for quantifying the loss of frozen ground, estimating the release of stored organic material, and can act as a basis for further related studies in Arctic coastal environments.

**Keywords:** permafrost; coastal erosion; deep learning; change vector analysis; Google Earth Engine; synthetic aperture RADAR





## 1. Introduction

Arctic environments are particularly sensitive to global warming, given that temperatures in the Arctic rise more than twice as rapidly compared to the global average [1]. This phenomenon is commonly labeled as Arctic amplification [2]. As a result, drastic changes can be observed everywhere in high-latitude regions. A widespread component of the Arctic, and one that is also heavily affected by climate change, is permafrost. More than 65% of the exposed land above 60°N and roughly one quarter of the total terrestrial area in the Northern Hemisphere is underlain by such permafrost [3,4]. It is defined as ground material that remains continuously frozen for two or more consecutive years [5]. However, the thermal state and distribution of the mentioned frozen ground are heavily impaired by the increase in ground temperature, which is reported for most regions across the permafrost domain [6–8].

The steadily deteriorating state of permafrost is reflected through the increasing erosion rates of Arctic coastlines during the last decades [9,10]. The average erosion rates have

hereby more than doubled for unlithified coasts within the permafrost domain of Siberia, Canada, and Alaska since the early 2000s compared to the late twentieth century [11]. The amplified erosion rates of permafrost coasts depend not only on the warming permafrost itself but also on other environmental factors and the interplay between them [9,12]. These factors include rising sea and air temperatures, higher storm frequencies, the increased duration of the open-water period, and the decreased extent of sea ice, amongst others [13]. A variety of processes and features connected to the increased erosion rates of the Arctic permafrost coasts are visualized in Figure 1. As a result, drastic changes in Arctic coastal environments can be observed. Eroding permafrost coastlines force changes in fish and wildlife habitats [12,14] and endanger human infrastructure and settlements [15,16]. Moreover, organic carbon content that was previously stored in the frozen masses of permafrost soils is released into the oceans [12,17]. According to current estimates, permafrost stores between 1460 and 1700 billion tonnes of organic material, a figure twice as large as the total amount of carbon in the atmosphere [7,18,19]. The carbon release from coastal erosion alone is hereby expected to increase up to 75% by the year 2100 [20].

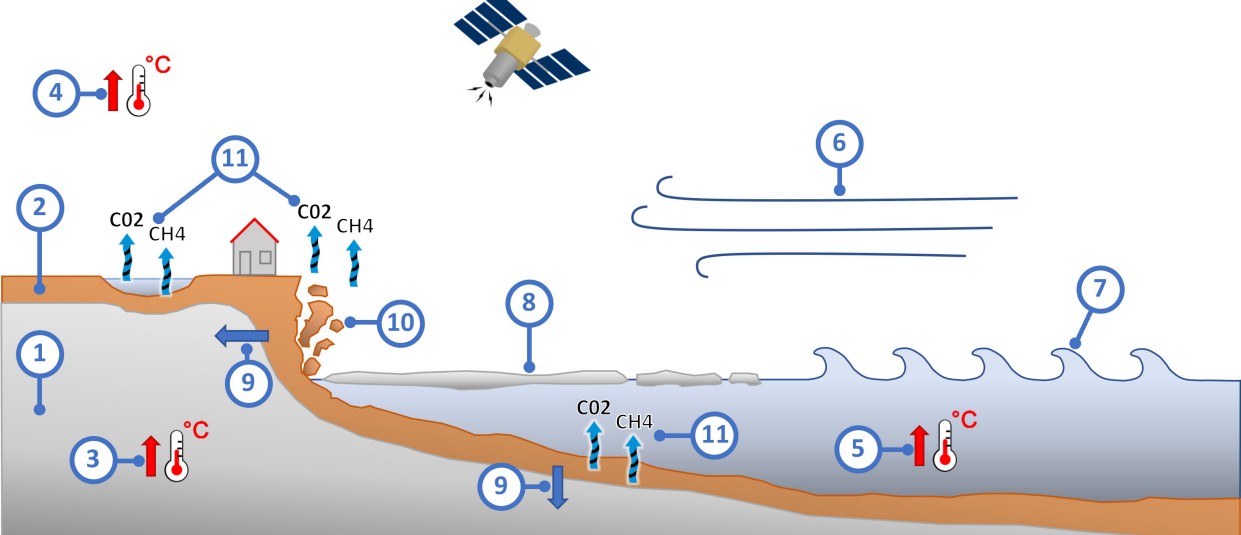

**Figure 1.** The info-graphic illustrates a variety of processes and features connected to increased erosion rates of Arctic permafrost coasts: (**1**) Permafrost, (**2**) unfrozen ground material, (**3**) increasing permafrost temperatures, (**4**) increasing air temperatures, (**5**), increasing sea temperatures, (**6**) increasing storminess, (**7**) wave action, (**8**) decreasing sea-ice extent, (**9**) decreasing permafrost extent, (**10**) coastal erosion, (**11**) release of organic carbon. Some symbols within the info-graphic were adopted or modified according to and courtesy of the Integration and Application Network, University of Maryland Center for Environmental Science [21].

Roughly one-third of Earth's coastlines are influenced by permafrost [10]. Therefore, it is crucial to have a good understanding of the current state of permafrost coasts and their erosion processes on large to circum-Arctic scales and with high detail in order to identify appropriate mitigation actions. For this purpose, satellite remote sensing is a powerful tool for spatially explicit, inexpensive, fast, and operational observations over large spatial scales and time. Nonetheless, satellite Earth observation analyses in Arctic regions remain challenging due to disadvantageous environmental conditions, such as low light intensities (including polar night), steep sun angles, and frequent cloud coverage [22,23]. Especially optical satellite imagery is heavily influenced by these environmental factors, which results in data gaps and therefore strongly limits the usability of this data type within the Arctic domain. Synthetic Aperture RADAR (SAR) data, on the other hand, is largely independent of sun illumination and weather conditions and has therefore the potential to overcome some of the limitations associated with optical imagery [24,25]. That said, SAR data also

come with challenges in the context of monitoring Arctic coastal erosion frequencies. In a recent study by Bartsch et al. [26], the applicability of three different wavelengths (X-, C-, and L-band) from three different satellite missions (TerraSAR-X, Sentinel-1 (S1), and ALOS PALSAR 1/2) were investigated for various study sites across the Arctic. While the application of SAR data for coastal erosion analysis was generally considered to be feasible across all wavelengths, the authors stress challenges in the form of ambiguous scattering behavior, issues with viewing geometries, and inconsistencies in data acquisition [26].

A first attempt at quantifying coastal erosion rates on a pan-Arctic scale was undertaken by Lantuit et al. [10] in the form of the Arctic Coastal Dynamics (ACD) database. The database provides a geomorphological classification for over 100,000 km of Arctic coastline into 1315 segments with information about, amongst others, the shore form, ground ice content, lithification stage, as well as average erosion rates per segment [10].

This study aims to explore new methodologies based on SAR satellite remote sensing data to further advance in quantifying Arctic coastal erosion rates with high spatial resolution and on a circum-Arctic scale. As a first step toward achieving this goal, an ongoing, spatio-temporally explicit, and easily reproducible coastal erosion and build-up product was created for ten test sites that are distributed across the Arctic. The annual median and standard deviation (sd) backscatter images derived from S1 and covering a total length of 1038 km of permafrost coastline were hereby generated. Lastly, nine different U-Net architectures were employed to compute a high-quality coastline product, which acted as a reference for the subsequent annual coastal erosion and build-up quantification based on Change Vector Analysis (CVA).

## 2. Study Area

A combined 1038 km of permafrost coastline and a total area of 19,275 km$^2$ divided into ten different regions across the Arctic were analyzed within the framework of this study (Table 1 and Figure 2). Areas in three different countries, the United States of America (USA), Canada, and Russia, were investigated. All of the selected regions feature significant erosion rates based on the ACD database by Lantuit et al. [10] and were therefore considered suitable study sites. Details about each area of interest (aoi), including the region's name, associated country, area, length of the present coastline, coordinates, and the relative orbit of available S1 data, are listed in Table 1. Figure 2 illustrates the spatial distribution of the study sites across the Northern Hemisphere.

**Table 1.** Regions of interest analyzed within the framework of this study. Information about the region's name, country, analyzed area, length of the present coastline and the center coordinates (coords.) per area of interest (aoi) are provided. Validation sites are marked in bold text.

| Number | Name | Country | Area | Coast Length | Center Coords. |
|---|---|---|---|---|---|
| **1** | **Corwin Bluffs** | **USA** | **1631 km$^2$** | **67.8 km** | **68.8°N; 165°W** |
| 2 | Drew Point–Cape Halkett | USA | 1335 km$^2$ | 142.4 km | 70.9°N; 153°W |
| 3 | Shoalwater Bay | Canada | 636 km$^2$ | 113.7 km | 68.8°N; 136.7°W |
| **4** | **Kolgujev** | **Russia** | **685 km$^2$** | **49.4 km** | **69°N; 48.2°E** |
| 5 | Sims Bay | Russia | 1280 km$^2$ | 97.2 km | 76.7°N; 109°E |
| **6** | **Mus-Khaya Cape–Mouth of Peshanaya** | **Russia** | **3454 km$^2$** | **170.2 km** | **73.6°N; 116°E** |
| 7 | Bykovsky Peninsula | Russia | 828 km$^2$ | 115 km | 71,9°N; 129.3°E |
| 8 | Muostakh Island | Russia | 212 km$^2$ | 15.7 km | 71.6°N; 130°E |
| 9 | Bezimyanniy Cape–Eastern Oyagoss Cape | Russia | 2748 km$^2$ | 109 km | 72.6°N; 144°E |
| 10 | Mouth of Kurdugina–Malyy Chukochiy Cape | Russia | 6476 km$^2$ | 158.4 km | 70.5°N; 159.8°E |

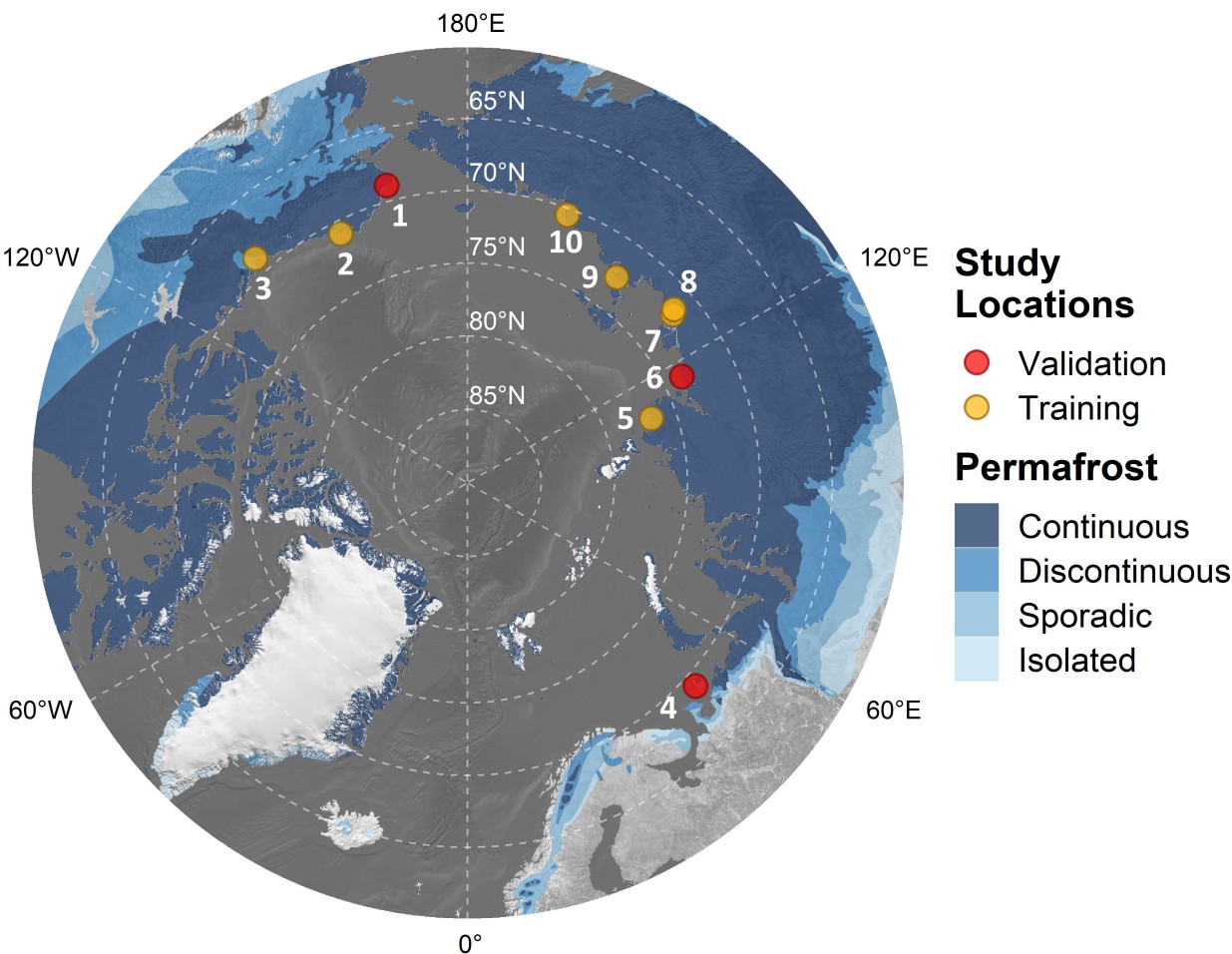

**Figure 2.** The study area distribution across the Northern Hemisphere together with the circum-Arctic permafrost map by Brown et al. [27] (bluish colored areas). The white bold numbers hereby correspond to the area of interest (aoi) number as listed in Table 1. A shaded relief by Natural Earth [28] was utilized as a background map. All data are visualized in a polar Lambert azimuthal equal-area projection.

## 3. Materials and Methods

In order to investigate erosion rates of Arctic permafrost coasts, the analysis for this study was split into three parts (Figure 3). In a first step, annual median backscatter and sd backscatter images were computed from S1 Ground Range Detected (GRD) scenes. Images for the years 2017 and 2020 were used throughout the study. The second part of the study was dedicated to computing a high-quality Arctic coastline product via Deep Learning (DL), using the pre-processed annual S1 composites as an input. In order to achieve this, nine different U-Net architectures were combined to generate the best possible output coastline. The mentioned coastline product could subsequently be employed in the third step as a reference to detect coastal changes. Erosion and build-up rates were hereby investigated through CVA. Again, the annual median and sd backscatter images were used as input. As a final step, average erosion and build-up rates were computed for 200 m segments along the predicted coastline. Details on the used satellite data, the pre-processing steps, the implemented algorithms for computing the reference coastline, and the quantification of coastal erosion and build-up rates, as visualized in Figure 3, are provided from here onward.

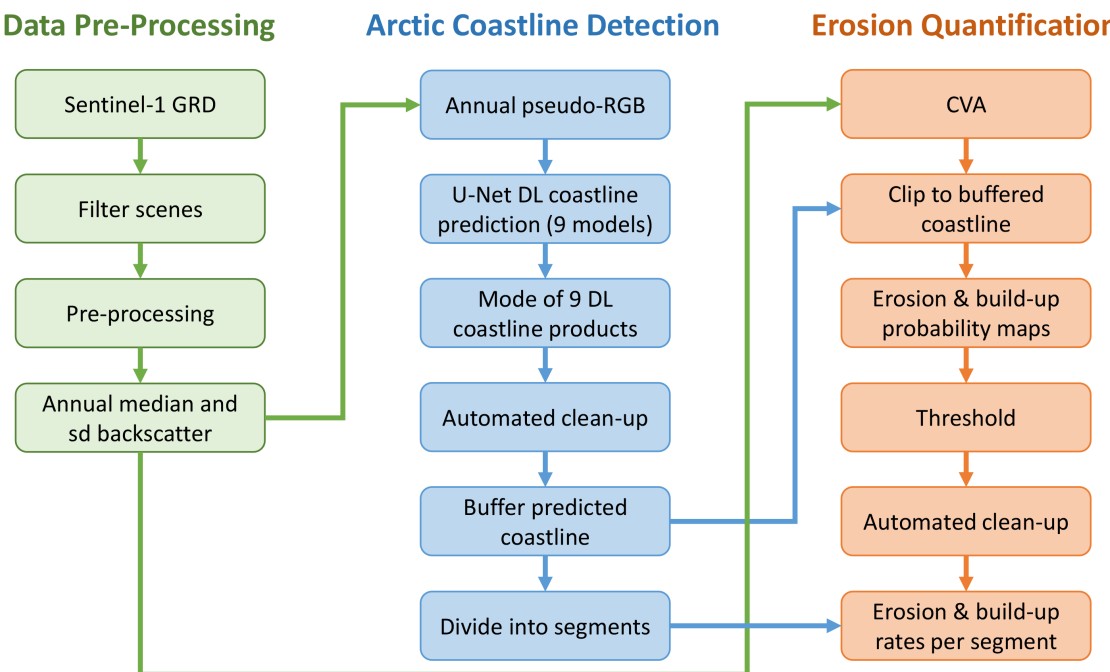

**Figure 3.** Flowchart of the study outline. The analysis was divided into three parts. In the first part, annual Sentinel-1 (S1) composites were computed. In the second part, a high-quality Arctic coastline product was generated via DL. The third part was dedicated to quantifying Arctic coastal erosion and build-up rates by using the previously computed coastline product as a reference. The following abbreviations are used throughout the info-graphic: Ground Range Detected (GRD); standard deviation (sd); Red Green Blue (RGB); DL; and CVA.

### 3.1. Satellite Remote Sensing Data and Pre-Processing

Openly available optical satellite imagery, such as provided by the Landsat legacy, allows for time series analysis since 1972 [29]. More recent satellite missions, for example, the Sentinel-2 (S2) mission, even provide publicly available optical data with a spatial resolution of up to 10 m and a revisit time of 5 days with both satellites S2 A/B combined [30]. However, sparse data availability for much of the Arctic region in conjunction with cloud contamination and the resulting data gaps heavily limit the usability of optical scenes for Arctic investigations [3,4,29]. SAR, on the other hand, has the ability to gather information independent of sun illumination or weather conditions (e.g., clouds) and has therefore the potential to overcome some limitations of optical imagery [24,25]. This study is dedicated to exploiting the continuous observation capabilities of SAR imagery in the form of S1 data in order to investigate Arctic permafrost coastal erosion rates.

S1 Level-1 GRD products in Interferometric Wide (IW) mode were hereby employed, which consist of focused SAR imagery that has been multi-looked, detected, and projected to ground range via an Earth ellipsoid model in combination with a Digital Elevation Model (DEM) [31]. The data were accessed through the cloud computing platform Google Earth Engine (GEE) and is provided with a spatial resolution of 10 m [32]. Imagery for the years 2017 and 2020 was used throughout this study. Further filtering steps included selecting the path direction (descending), only using images taken during the summer months (June–September) in order to avoid sea-ice contamination, and lastly, reducing the remaining scenes to only have images from the most frequent relative orbit per aoi. Table 2 lists the number of available scenes per aoi and year after the filtering approach. S1 GRD data in GEE are provided as backscatter coefficient sigma nought ($\sigma^0$) in the unit decibel (dB) [32]. The data were subsequently converted from dB to natural scale using Equation (1). In order to reduce the amount of speckle, a median Moving Window (MV) was applied on each SAR scene. Afterward, the imagery was converted back to dB via Equation (2). As a next step, annual median backscatter and annual sd backscatter images

were computed for each polarization (VV and VH). The median was chosen as a statistical parameter because it is more robust to outliers compared to the arithmetic mean [33]. Working on annual composites instead of single scenes significantly reduced the speckle effect and the geolocation uncertainty [34,35]. Furthermore, having annual median and sd backscatter composites allowed for the creation of annual pseudo-Red Green Blue (RGB) images, for which median backscatter VV, VH, and sd VV composites were combined. Each of the annual composites was hereby normalized by applying a linear stretch between the 2nd and 98th percentile. Figure 4 visualizes an exemplary pseudo-RGB image and the annual composites it is made from. Lastly, all data were re-projected to a polar Lambert azimuthal equal-area projection for further processing. All data access, filtering, and pre-processing were conducted in GEE.

$$\sigma^0 = 10^{\frac{\sigma^0(db)}{10}} \tag{1}$$

$$\sigma^0(dB) = 10 * \log_{10} \sigma^0 \tag{2}$$

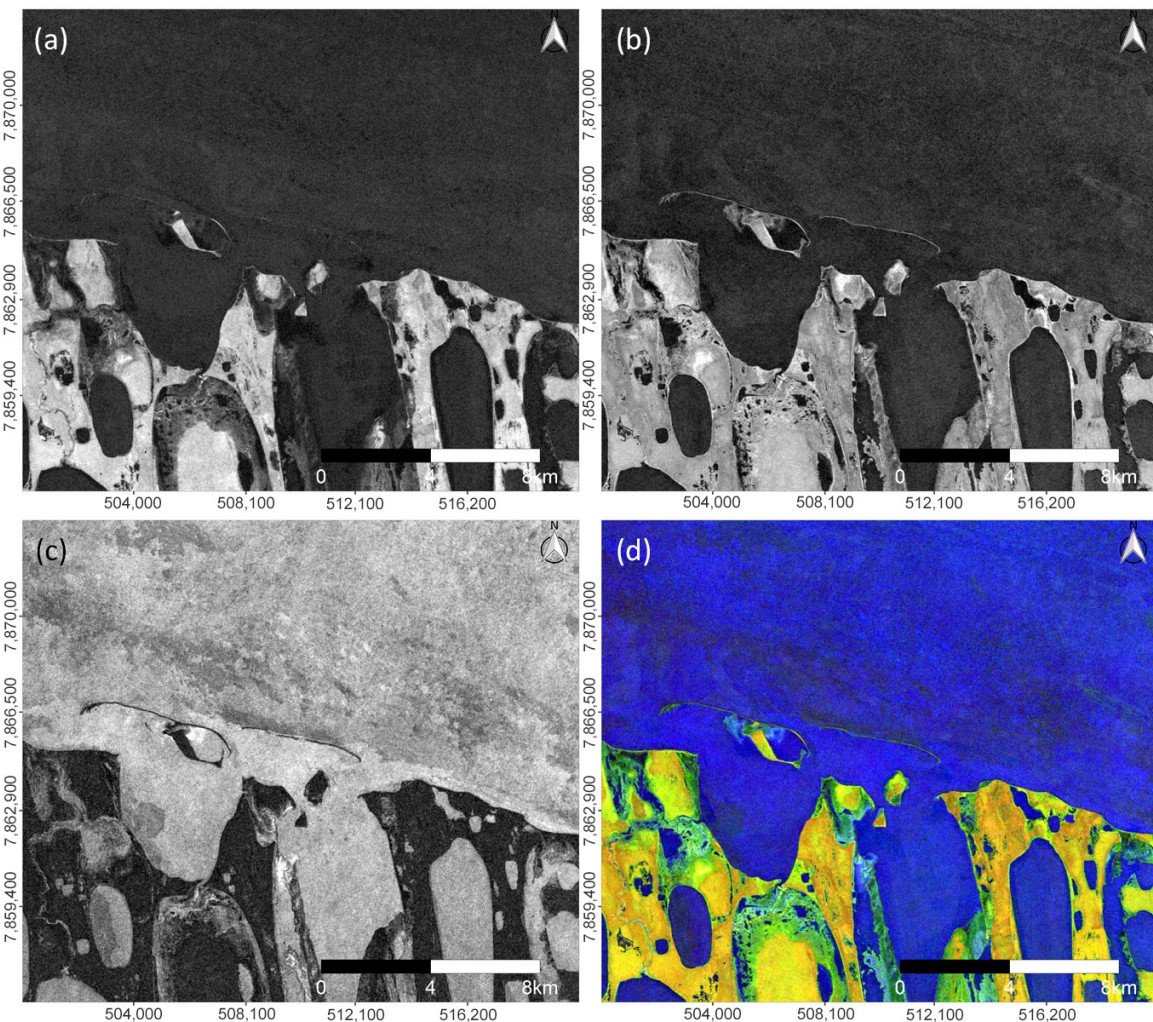

**Figure 4.** Subsection of the study area Cape Halkett in the United States of America (USA) (aoi 02) visualized by (**a**) annual median VH backscatter, (**b**) annual median VV backscatter, (**c**) annual sd VV backscatter, and (**d**) a pseudo-RGB composite of (**a–c**) based on S1 GRD scenes from June to September 2020.

**Table 2.** Number of available Sentinel-1 (S1) Ground Range Detected (GRD) Interferometric Wide (IW) scenes per year and path direction after filtering the data to summer months (June–September) and further only selecting images from the most frequent relative orbit per aoi. Validation sites are marked in bold text. The minus symbol "-" represents no available scenes.

| AOI | Year | No. of Scenes | Rel. Orbit |
|---|---|---|---|
| **1** | **2017** | **10 (Desc.); 9 (Asc.)** | **88 (Desc.); 153 (Asc.)** |
| | **2020** | **9 (Desc.); 9 (Asc.)** | **88 (Desc.); 153 (Asc.)** |
| 2 | 2017 | 10 (Desc.); 9 (Asc.) | 73 (Desc.); 94 (Asc.) |
| | 2020 | 10 (Desc.); 10 (Asc.) | 73 (Desc.); 94 (Asc.) |
| 3 | 2017 | 10 (Desc.); 5 (Asc.) | 116 (Desc.); 108 (Asc.) |
| | 2020 | 10 (Desc.); 10 (Asc.) | 116 (Desc.); 108 (Asc.) |
| **4** | **2017** | **10 (Desc.); - (Asc.)** | **123 (Desc.); - (Asc.)** |
| | **2020** | **7 (Desc.); - (Asc.)** | **123 (Desc.); - (Asc.)** |
| 5 | 2017 | 10 (Desc.); - (Asc.) | 48 (Desc.); - (Asc.) |
| | 2020 | 5 (Desc.); - (Asc.) | 48 (Desc.); - (Asc.) |
| **6** | **2017** | **10 (Desc.); - (Asc.)** | **135 (Desc.); - (Asc.)** |
| | **2020** | **10 (Desc.); - (Asc.)** | **135 (Desc.); - (Asc.)** |
| 7 | 2017 | 10 (Desc.); - (Asc.) | 149 (Desc.); - (Asc.) |
| | 2020 | 8 (Desc.); - (Asc.) | 149 (Desc.); - (Asc.) |
| 8 | 2017 | 10 (Desc.); - (Asc.) | 149 (Desc.); - (Asc.) |
| | 2020 | 8 (Desc.); - (Asc.) | 149 (Desc.); - (Asc.) |
| 9 | 2017 | 10 (Desc.); - (Asc.) | 61 (Desc.); - (Asc.) |
| | 2020 | 9 (Desc.); - (Asc.) | 61 (Desc.); - (Asc.) |
| 10 | 2017 | 10 (Desc.); - (Asc.) | 31 (Desc.); - (Asc.) |
| | 2020 | 10 (Desc.); - (Asc.) | 31 (Desc.); - (Asc.) |

*3.2. Deep Learning Coastline Detection*

DL Convolutional Neural Networks (CNNs) have become increasingly popular in recent years. Especially because, in many cases, a better performance of CNNs over conventional Machine Learning (ML) classifiers, such as support vector machine or random forest, are reported [36,37]. In particular, better segmentation performance between land and water was observed for CNNs over traditional image processing, making it an attractive tool for coastline identification [37,38]. One major goal of this study is to exploit the segmentation capabilities of DL in order to create a high-quality Arctic coastline product, which will act as a reference for the analysis of coastal erosion and build-up. For this purpose, a CNN-based U-Net architecture was employed, which proved to be highly capable in the context of detecting coastlines using SAR imagery [39–42]. A brief overview of the CNN-based U-Net structure and the hyper-parameters used within this study will be given here.

A U-Net architecture consists of a contracting path (also called encoder) that extracts the context of a scene and a symmetric expanding path (also called decoder), which allows for the exact localization of the detected features [43]. During the contracting path, a set of filters (also called kernels) that highlight certain features are applied to an input image. The introduced kernels with a size of $3 \times 3$ run across the image in an MV fashion with a stride of 1 and extract the dot product for a given position. Padding is applied, which adds a layer of zeros around the frame of the input image in order to avoid shrinking in x- and y-dimensions during the convolution process. The output product is a feature map, on which a Rectified Linear Unit (ReLU) activation function (Equation (3)) is applied. The ReLU function replaces negative values within the feature map with 0. Subsequently, MaxPooling is executed on the feature map which describes the application of MV that selects the maximum value for a given location. A window size of $2 \times 2$ and a stride of 2 is hereby used. This process reduces the size of the feature map and therefore lowers the computational cost. The described procedure is repeated several times while an increasing number of filters are applied in each step. This consequently reduces the size of the output

in x- and y-dimensions while at the same time the number of feature maps increases. As the information becomes denser, the localization of the same information is getting lost. During the expanding path, this spatial information is restored by up-sampling the feature maps and connecting them with the set of corresponding feature maps from the contracting path. This process is repeated until the final set of feature maps have the same x- and y-dimensions as the input file. A final convolution in combination with a Sigmoid activation function (Equation (4)) converts the up-sampled feature maps into a probability map with pixel values ranging between 0 and 1. Figure 5 visualizes the schematic structure of a U-Net. A more detailed description of the framework for the U-Net model can be found in the original paper by Ronneberger et al. [43].

$$y = max(0, x) \tag{3}$$

$$\sigma(x) = \frac{1}{1 + e^{-x}} \tag{4}$$

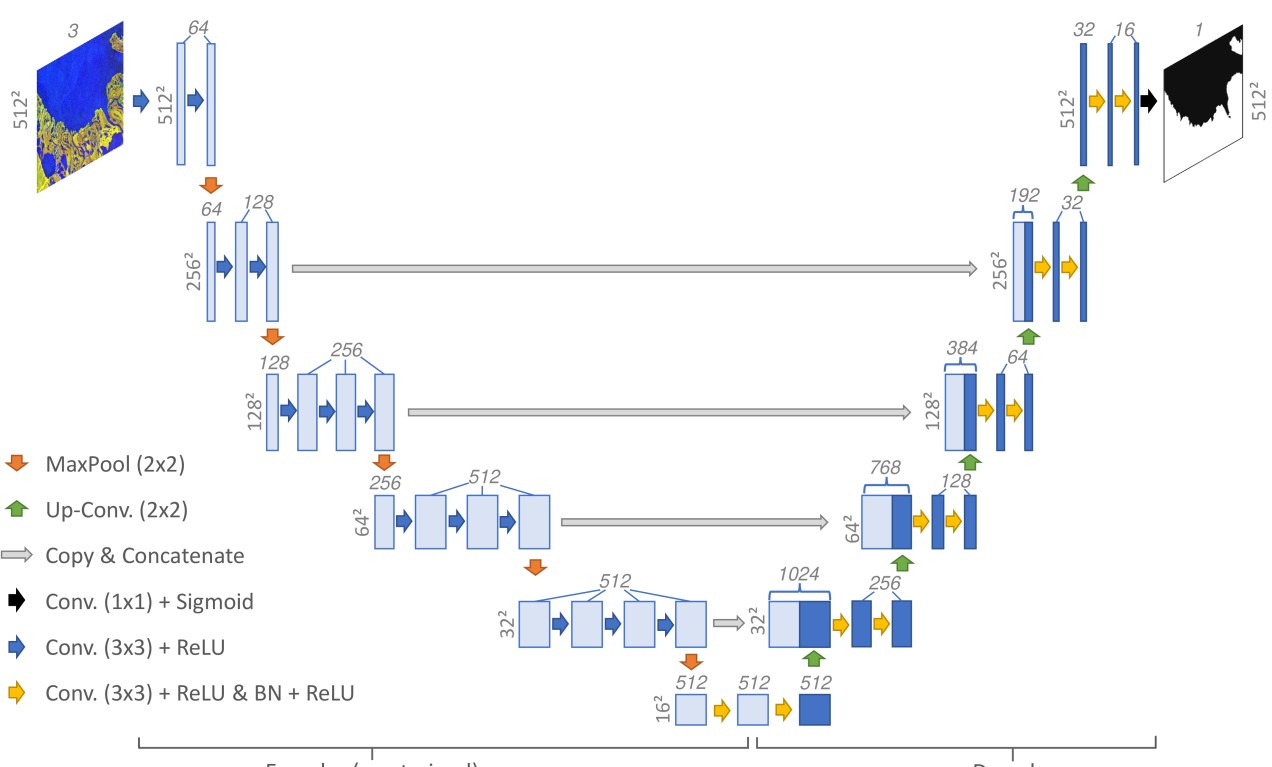

**Figure 5.** Schematic visualization of a VGG16 U-Net architecture. The vertical numbers at the side of each feature block represent the x–y-dimensions. The numbers in italics above each block represent the amount of feature maps (layers). Feature maps inside the down-sampling path of the (pre-trained) encoder are displayed in light blue color, whereas feature maps within the up-sampling path of the decoder are visualized in dark blue. The following abbreviations are used: Convolution (Conv), Batch Normalization (BN), and Rectified Linear Unit (ReLU).

The mentioned kernels have random initialized weights that are updated during the training of the model. Different hyper-parameter settings were tested for the best possible performance. All input scenes featured 512 pixels for both x- and y-dimensions as well as three layers (VH median, VV median, VV sd). For model training, a batch size of 8, a Root Mean Square Propagation (RMSprop) optimizer with a learning rate of 0.001, a binary cross-entropy loss function, and binary accuracy as an accuracy metric were used. As shown in Equation (5), the binary cross-entropy loss function (also called log loss) takes the negative log of the predicted probability for a given class [44]. Therefore, the loss value

increases exponentially the closer the predicted probability for the actual class is to zero. As a result, big differences between the predicted probability and the correct probability are penalized with a high loss value.

$$CE = -\frac{1}{n}\sum_{i=1}^{n} y_i \cdot \log(p_i) + (1 - y_i) \cdot \log(1 - p_i) \qquad (5)$$

where

| | | |
|---|---|---|
| $CE$ | = | binary cross entropy; |
| $n$ | = | total number of observations; |
| $i$ | = | current observation; |
| $y$ | = | current label $\in \{0,1\}$; |
| $p$ | = | probability of belonging to label 1. |

The U-Net model was applied to perform a segmentation between sea and land areas (including inland lakes and rivers). In order to increase the quality of the resulting segmentation maps, a total of nine different U-Net architecture types were trained, and their results were combined within the context of this study. The following architectures were used: VGG16 [45], VGG19 [45], ResNet34 [46], ResNet50 [46], Inception v3 [47], Inception-ResNet v2 [48], ResNeXt [49], DenseNet121 [50], and SE-ResNeXt50 [51]. Each model was available with pre-trained encoder weights based on the ImageNet database, which consists of roughly 14 million images categorized in more than 20,000 classes [52]. In addition, another 32,606 normalized and manually labeled S1 Pseudo-RGB clips covering the seven training sites (Table 1) were used for training and a further 16,490 separate scenes from the independent validation sites for validation. Augmentation in the form of flipping and rotating images by 90, 180, and 270 degrees was applied in order to increase the amount of training data. The validation images originate from the three independent validation sites as visualized in Figure 2 and listed in Table 1. A manually digitized coastline based on S1, S2, and high-resolution Google Earth imagery served hereby as a reference.

Each architecture was trained for 30 epochs, and the epoch with the highest validation accuracy was used as the representative output. The output was a probability map with pixel values ranging between 0 and 1. A threshold of 0.5 was applied to differentiate between the two classes, sea and land. Lastly, the mode of the nine separate segmentation maps was computed to identify the most probable class (terrestrial area, including inland lakes and rivers vs. sea) per pixel. Post-processing included the removal of objects smaller than 0.2 km$^2$ as well as closing holes smaller than 3 km$^2$. The accuracy of the predicted coastline was compared to other freely available and circum-Arctic coastline products, including OpenStreetMap (OSM) [53], the Global Self-consistent, Hierarchical, High-resolution Geography Database (GSHHG) product [54], and the Circumpolar Arctic Vegetation Map (CAVM) [55].

### 3.3. Arctic Coastal Erosion Quantification via Change Vector Analysis

CVA is a commonly applied tool for detecting land-cover change via satellite remote sensing imagery. In contrast to traditional post-classification change detection, which extracts information of change via the comparison of two individual classifications, CVA applies radiometric comparison to extract change information and therefore avoiding an accumulation of errors from separate input classifications [56]. Furthermore, CVA allows for the identification of both the direction and magnitude of change [57]. The previously calculated annual median backscatter and annual sd backscatter in VV polarization were used as inputs for the CVA computation. As visualized in Figure 4, on average, the median backscatter proved to be higher over land and lower over water, whereas the sd backscatter was higher over water and lower over land. Thus, a change from a high median backscatter to high sd backscatter can be interpreted as a change from land to water. Respectively, a change from high sd backscatter to a high median backscatter can be interpreted as a change from water to land. This build-up is realized by constructive waves that swash ma-

rine sediments toward the coast and, at the same time, have a relatively small backwash [58]. The magnitude is based on the Euclidean distance between two positions of the same pixel from different data in a two-dimensional Euclidean plane (Equation (6)) [59]. A one-sided buffer of 200 m directed toward the sea was applied on the previously derived high-quality coastline product. The buffered coastline was used as a mask to clip the magnitude of change maps. In addition, a 50 m buffer directed inland was applied to the coastline product in order to account for any potential inaccuracies of the predicted coastline. As a next step, the CVA-based magnitude of change maps were normalized to generate probability maps of change (0–1). The analyzed S1 scenes, together with satellite data from Landsat 8 and S2 for the years 2017 and 2020, as well as further high-resolution imagery from Google Earth were used as a reference to identify the most suitable threshold values for detecting the actual change in the form of coastal erosion and build-up. The derived threshold values were applied to the probability maps and a mode-MV was employed to reduce the amount of leftover noise to a minimum. As a final step, average erosion and build-up rates for 200 m segments along the coastline were calculated. While several studies successfully applied CVA in the context of land-cover change analysis [60–67], to the best of our knowledge, the combination of CVA and S1 has not been employed before in the context of quantifying erosion rates of permafrost coasts.

$$d_{(x,y)} = \sqrt{(x_i - y_i)^2 + (x_j + y_j)^2} \tag{6}$$

where

$d$ = euclidean distance;
$x$ = Date 1 (in this study: year 2017);
$y$ = Date 2 (in this study: year 2020);
$i$ = Band 1 (in this study: VV standard deviation backscatter);
$j$ = Band 2 (in this study: VV median backscatter).

*3.4. Validation*

The ten investigated areas as listed in Table 1 and Figure 2 were divided into seven training and three validation sites for the generation of a high-quality coastline product via DL. Splitting the data this way avoided spatial auto-correlation between the training and test datasets while ensuring that accuracy metrics are not inflated. In order to train the U-Net models and for product quality assessment of the DL-based coastline predictions, 1038 km of manually digitized reference coastlines for the year 2020 were created for all ten regions of interest (Figure 6). This reference information is based on S1, S2, and high-resolution Google Earth imagery. The deviation of the predicted coastline to the reference coastline served hereby as a means to quantify the accuracy of the final coastline product. In addition, the accuracy of the final binary classification maps per aoi, after combining the results of the nine different architectures and post-processing, was assessed within a buffer of 500 m around the manually digitized coastline. As the result of the DL prediction is a binary map with two classes, overall accuracy can be expected to be high for an entire scene. However, as the focus of this study lies on the transition between sea and land area, the accuracy assessment was concentrated around the coastline to provide more meaningful insight into the quality of the generated products. For this purpose, common performance measures, including overall accuracy, precision, recall, and the $F_1$-score, are derived for the binary classification maps per aoi. Precision captures hereby the amount of correct positive predictions in relation to the total amount of positive predictions and is, therefore, a suitable measure for false positives [68]. Recall, on the other hand, sets the correct positive predictions in relation to the actual total amount of positives and provides a good measure for capturing real positives [69]. The $F_1$-score is a balance of both by providing the harmonic mean between precision and recall [70].

Finding a suitable threshold is essential when converting the CVA-based probability of change maps into binary maps for coastal erosion and build-up areas. S1 scenes in

combination with high-resolution imagery from Google Earth and further satellite data from Landsat 8 and S2 were used as a reference to manually digitize coastlines for both years 2017 and 2020. The difference between the two manually digitized reference lines was subsequently compared to the CVA-based change maps after applying different thresholds. Areas with significant erosion rates (e.g., Drew Point, Alaska) (Figure 7) but also areas with no erosion due to lithified coasts (e.g., eastern Kola peninsula) were hereby used as reference sites. This was undertaken to identify the most suitable threshold values for erosion and build-up quantification and therefore avoid over- or underestimations in coastal change rates while at the same time keeping the amount of noise to a minimum. The deviation of the predicted change to the actual coastal change acted hereby as a metric for the quality assessment of the coastal change products.

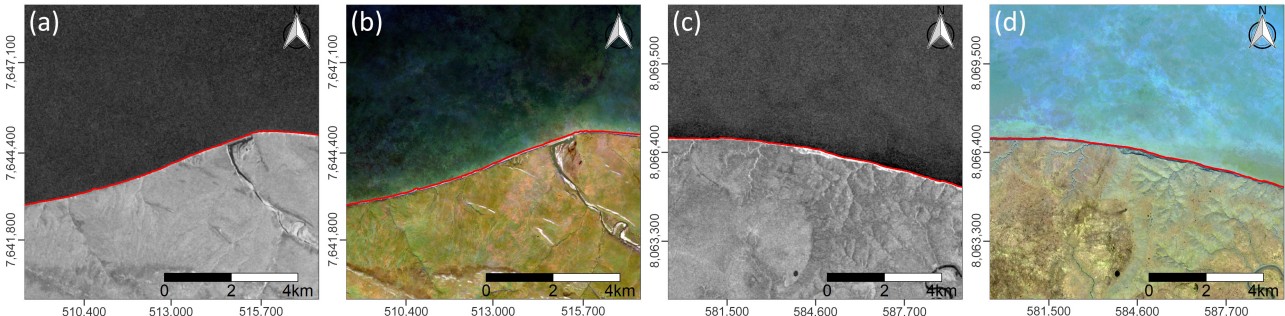

**Figure 6.** Subsections of the two regions of interest, Corwin Bluffs in the USA (aoi 01) (**a**,**b**) and Bezimyanniy Cape in Russia (aoi 09) (**c**,**d**). Median images (months June–September) for the year 2020 from S1 in VV polarization (**a**,**c**) and Sentinel-2 (S2) in RGB (**b**,**d**) are shown together with the manually digitized reference coastline (red line).

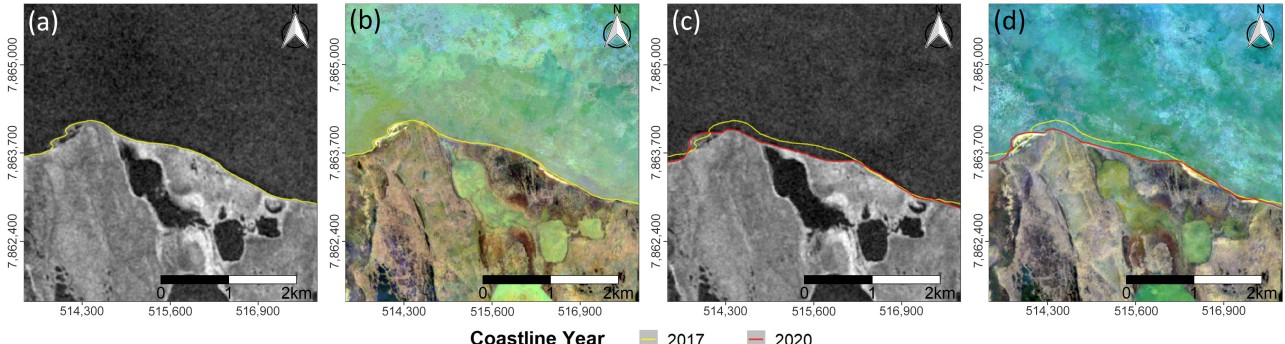

**Figure 7.** Subsections of Cape Halkett in the USA (aoi 02) in the year 2017 (**a**,**b**) and 2020 (**c**,**d**). Median images (months June–September) from S1 in VV polarization (**a**,**c**) and S2 in RGB (**b**,**d**) are shown together with the manually digitized reference coastlines for the years 2017 (yellow line) and 2020 (red line).

## 4. Results

### 4.1. Deep Learning Coastline Product

The training accuracy of the applied U-Net architectures ranged from 0.9977 (ResNet50 and DenseNet121) to 0.9998 (SE-ResNeXt50) with an average training accuracy of 0.9989. Similarly high validation accuracies between 0.9957 (Inception v3) and 0.998 (VGG19 and ResNet34) with an average validation accuracy of 0.9973 were achieved. The respective loss rates ranged from 0.0005 (SE-ResNeXt50) to 0.0091 (DenseNet121) with an average loss rate of 0.0039 in the case of the training data. For the validation data, loss rates between 0.0075 (VGG19) and 0.0167 (Inception v3) as well as an average loss rate of 0.0106 were observed. The details about the accuracy and loss rates in combination with the associated number of epochs for each model are listed in Table A1.

The accuracy metrics derived within a 500 m buffer around the reference coastline for the final combined binary classification maps after post-processing feature slightly lower yet still generally high overall accuracy scores. Values between 0.965 and 0.99 could be observed. The overall accuracy across all aois within the mentioned buffer was 0.974. The precision measures for the terrestrial areas (including inland lakes and rivers) were within the range of 0.945–0.995 with an average precision of 0.974. The precision values for sea area ranged between 0.953 and 0.999 with an average precision of 0.974, as well. The recall measures for the terrestrial areas ranged from 0.944 to 0.998, and an average recall of 0.972 was observed. For the sea areas, the recall values ranged between 0.936 and 0.995 with an average recall of 0.976. Lastly, the $F_1$-scores proved to be between 0.964 and 0.991 for the terrestrial areas with an average $F_1$-score of 0.973. The $F_1$ measures for the sea areas ranged between 0.963 and 0.993 with an average $F_1$-score of 0.975.

The average overall accuracy was 0.973 for the training aois and 0.975 for the validation sites. The average precision in the case of the training areas proved to be 0.971 for the terrestrial areas and 0.976 for the sea areas. The average precision of the test sites was revealed to be 0.982 for the terrestrial areas and 0.968 for the sea areas. The average recall values were 0.975 for terrestrial and 0.973 for sea within the training areas, and 0.967 for the terrestrial areas and 0.982 for the sea areas within the validation sites. Finally, the average $F_1$-scores within the training sites proved to be 0.973 for the terrestrial areas and 0.975 for the sea areas. $F_1$-scores of 0.974 for the terrestrial areas and 0.975 for the sea areas were observed for the test sites. The details about the overall accuracy, recall, precision, and $F_1$-scores per aoi and class are listed in Table 3.

**Table 3.** Accuracy statistics within a 500 m buffer around the manually digitized reference coastline for the final combined binary classification map after post-processing. Precision, recall, and $F_1$-scores are given for both classes, terrestrial area (including inland lakes and rivers) and sea, and each aoi. Validation sites are marked in bold text. Accuracy measures are rounded to the third decimal place.

| AOI | Overall Acc. | Label | Precision | Recall | F1 |
| --- | --- | --- | --- | --- | --- |
| **1** | **0.992** | **Terrestrial** | **0.995** | **0.988** | **0.991** |
|  |  | **Sea** | **0.988** | **0.995** | **0.992** |
| 2 | 0.972 | Terrestrial | 0.99 | 0.944 | 0.967 |
|  |  | Sea | 0.958 | 0.993 | 0.975 |
| 3 | 0.967 | Terrestrial | 0.947 | 0.993 | 0.97 |
|  |  | Sea | 0.992 | 0.936 | 0.963 |
| **4** | **0.982** | **Terrestrial** | **0.995** | **0.969** | **0.982** |
|  |  | **Sea** | **0.971** | **0.995** | **0.983** |
| 5 | 0.969 | Terrestrial | 0.987 | 0.948 | 0.967 |
|  |  | Sea | 0.953 | 0.988 | 0.97 |
| **6** | **0.965** | **Terrestrial** | **0.972** | **0.957** | **0.964** |
|  |  | **Sea** | **0.958** | **0.973** | **0.965** |
| 7 | 0.977 | Terrestrial | 0.977 | 0.973 | 0.975 |
|  |  | Sea | 0.977 | 0.981 | 0.979 |
| 8 | 0.99 | Terrestrial | 0.965 | 0.998 | 0.981 |
|  |  | Sea | 0.999 | 0.987 | 0.993 |
| 9 | 0.964 | Terrestrial | 0.945 | 0.983 | 0.964 |
|  |  | Sea | 0.983 | 0.945 | 0.963 |
| 10 | 0.986 | Terrestrial | 0.977 | 0.996 | 0.986 |
|  |  | Sea | 0.996 | 0.976 | 0.986 |

Figure 8 illustrates the annual S1 pseudo-RGB scenes, the respective DL-based binary classification maps as well as the derived coastlines for four different regions across the Arctic. A comparison of the predicted coastline to the reference coastline for Drew Point–Cape Halkett in the United States is shown in Figure 9. As reflected in the statistics listed

in Tables A1 and 3, the predicted coastline runs closely to the reference line. An elongated embankment near the coast, that is not present in the predicted dataset, is hereby an exception.

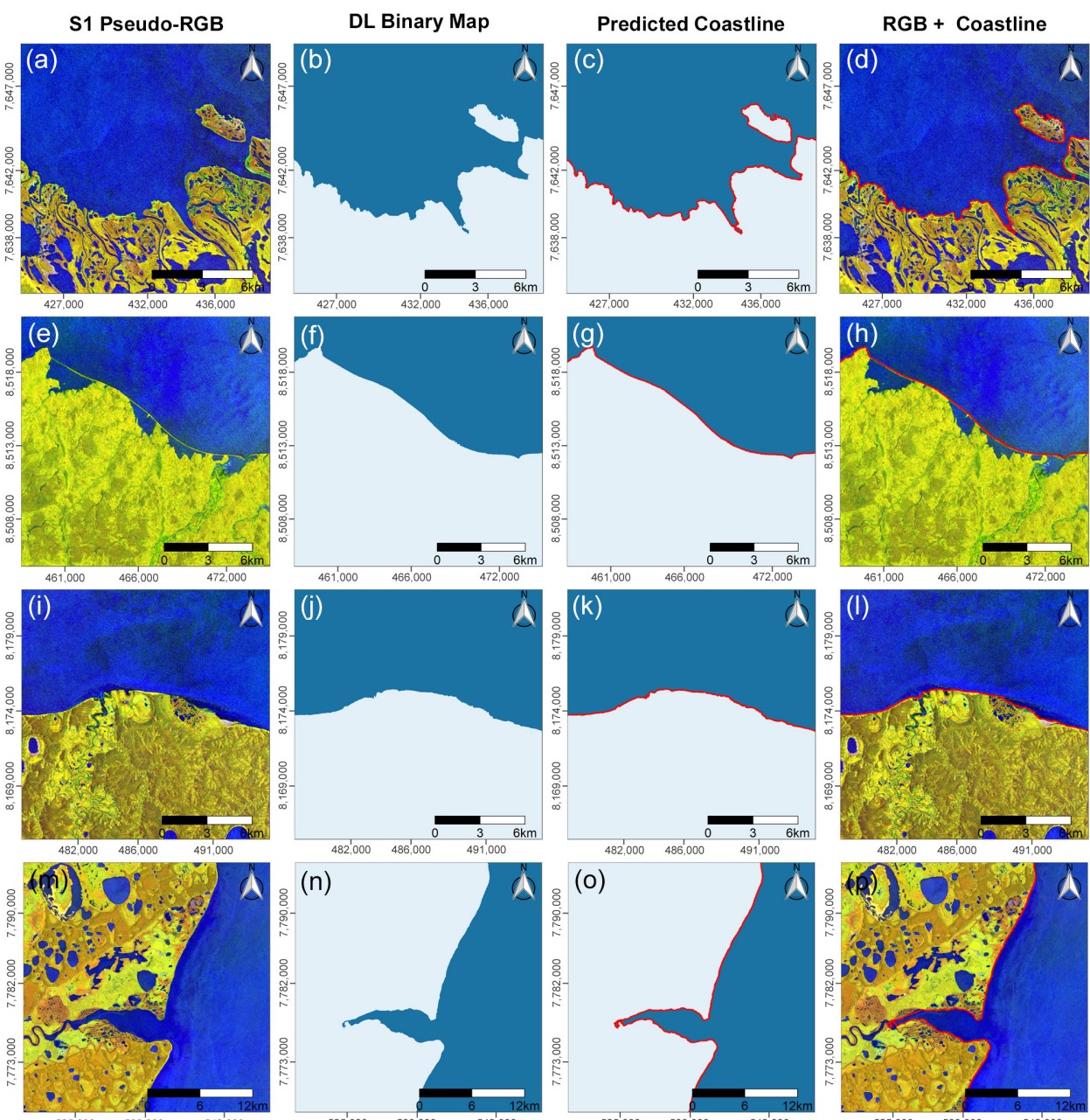

**Figure 8.** Subsets of annual S1 pseudo-RGB images, the resulting DL-based binary classification maps, and the derived coastlines for four different regions, including Shoalwater Bay in Canada (aoi 03) (**a–d**), Sims Bay in Russia (aoi 05) (**e–h**), Mus-Khaya Cape–Mouth of Peshanaya in Russia (aoi 06) (**i–l**), and Mouth of Kurdugina–Malyy Chukochiy Cape in Russia (aoi 10) (**m–p**).

The final coastline product has an average deviation of ±28 m to the reference coastline. A minimum distance of 0 m, a maximum distance of ±1905.6 m, and an sd of 110.8 m to the reference data were observed. The median proved to be ±6.9 m and the 2nd and 98th percentiles are ±0.2 m and ±378 m, respectively. The DL-based coastline outperformed other circum-Arctic and freely available datasets such as coastline products accessible through OSM [53], the GSHHG shoreline product [54], and the CAVM [55] in terms of accuracy (Table 4). Out of the three mentioned products, OSM performed best with an

average distance of ±331 m and a median distance of ±40.1 m to the reference data. However, OSM also featured the strongest maximum deviation of ±5525.6 m, the highest sd of 768.7 m, and the highest value for the 98th percentile of ±3215.6 m to the reference coastline. The GSHHG featured the second-best accuracy in terms of the average and median distance to the reference coastline of ±563 and ±386.8 m, respectively. The overall strongest deviation to the reference was observed for the CAVM coastline product with an average of ±707.7 m and a median of ±584.6 m.

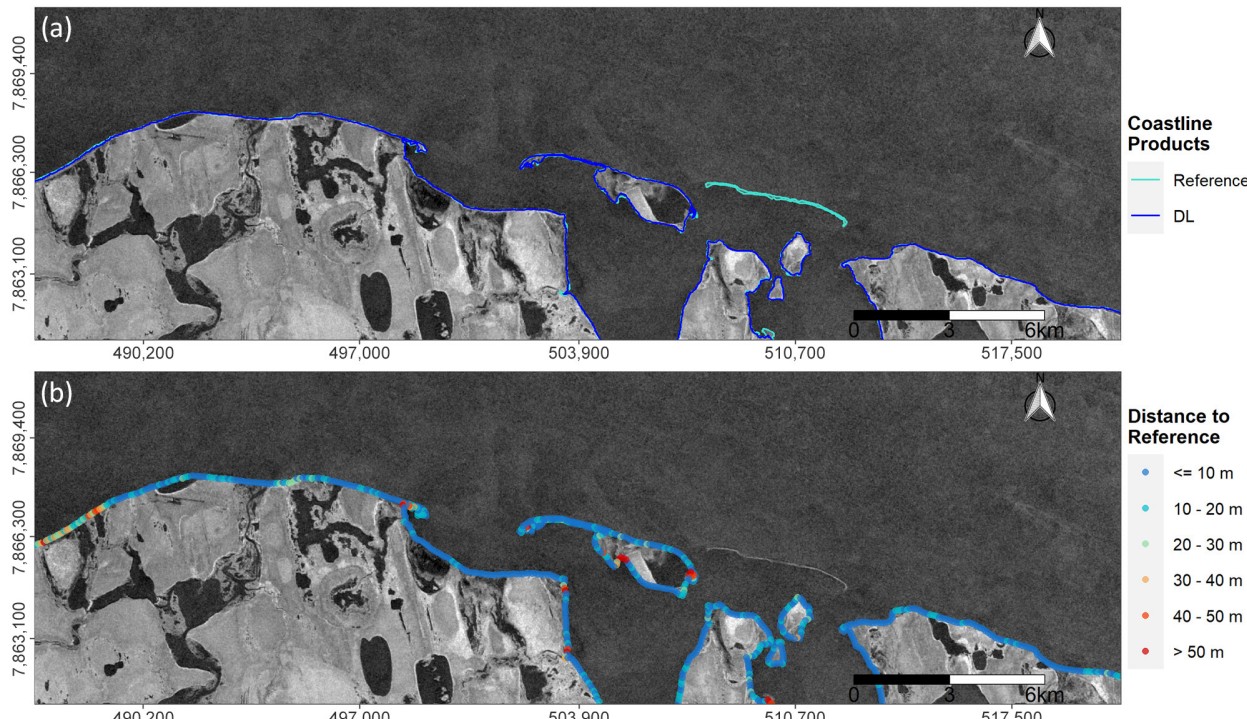

**Figure 9.** Comparison between the location of the reference coastline (turquoise line) and the DL-based predicted coastline (dark blue line) (**a**) as well as the distance of the predicted line to the reference data (**b**) for Drew Point–Cape Halkett, USA (aoi 02). An S1 annual median backscatter image for the year 2020 (months June–September) in VV polarization is used as a background image.

**Table 4.** Comparing the distance of different coastline products to the reference coastline. Four different coastline products are listed. The Deep Learning (DL)-based coastline product that was computed within the framework of this study as well as the existing coastline products, Circumpolar Arctic Vegetation Map (CAVM), OpenStreetMap (OSM), and the Global Self-consistent, Hierarchical, High-resolution Geography Database (GSHHG). Various statistics are listed, including the arithmetic mean, minimum (Min), maximum (Max), standard deviation (SD), median, and the 2nd (P02) and 98th (P98) percentile.

| Name | Mean | Min | Max | SD | Median | P02 | P98 |
|------|------|-----|-----|-----|--------|-----|-----|
| DL | 28 m | 0 m | 1905.6 m | 110.8 m | 6.9 m | 0.2 m | 378 m |
| OSM | 331 m | 0 m | 5525.6 m | 768.7 m | 40.1 m | 1.1 m | 3215.6 m |
| GSHHG | 563 m | 0.2 m | 5098.4 m | 614.4 m | 386.8 m | 12.9 m | 2527.6 m |
| CAVM | 707.2 m | 0 m | 3773.8 m | 642.6 m | 584.6 m | 12.9 m | 2828.3 m |

*4.2. Coastal Erosion and Build-Up Rates*

Comparing the CVA probability of the change maps to the manually digitized coastlines for both years, 2017 and 2020, revealed the most suitable thresholds to be 0.35 for erosion and 0.6 for build-up detection. The average deviation of the thresholded change

to the reference was −10.3 m. An sd of 12.9 m and an $r^2$ value of 0.92 could be observed between the predicted coastal change and the reference data.

Figure 10 visualizes the results of the CVA analysis for the two subsets of the investigation regions, Drew Point–Cape Halkett, USA (aoi 02), and Shoalwater Bay, Canada (aoi 03). The probability maps of change are shown in Figure 10a,c. Next to the actual coastal change, some noise can be observed over the water areas in the form of the low probability of change pixels. By applying the previously derived thresholds of 0.35 for erosion and 0.6 for build-up change detection, the majority of the in-water-noise could be removed. The resulting change maps feature areas of erosion and build-up with a low amount of noise (Figure 10b,d).

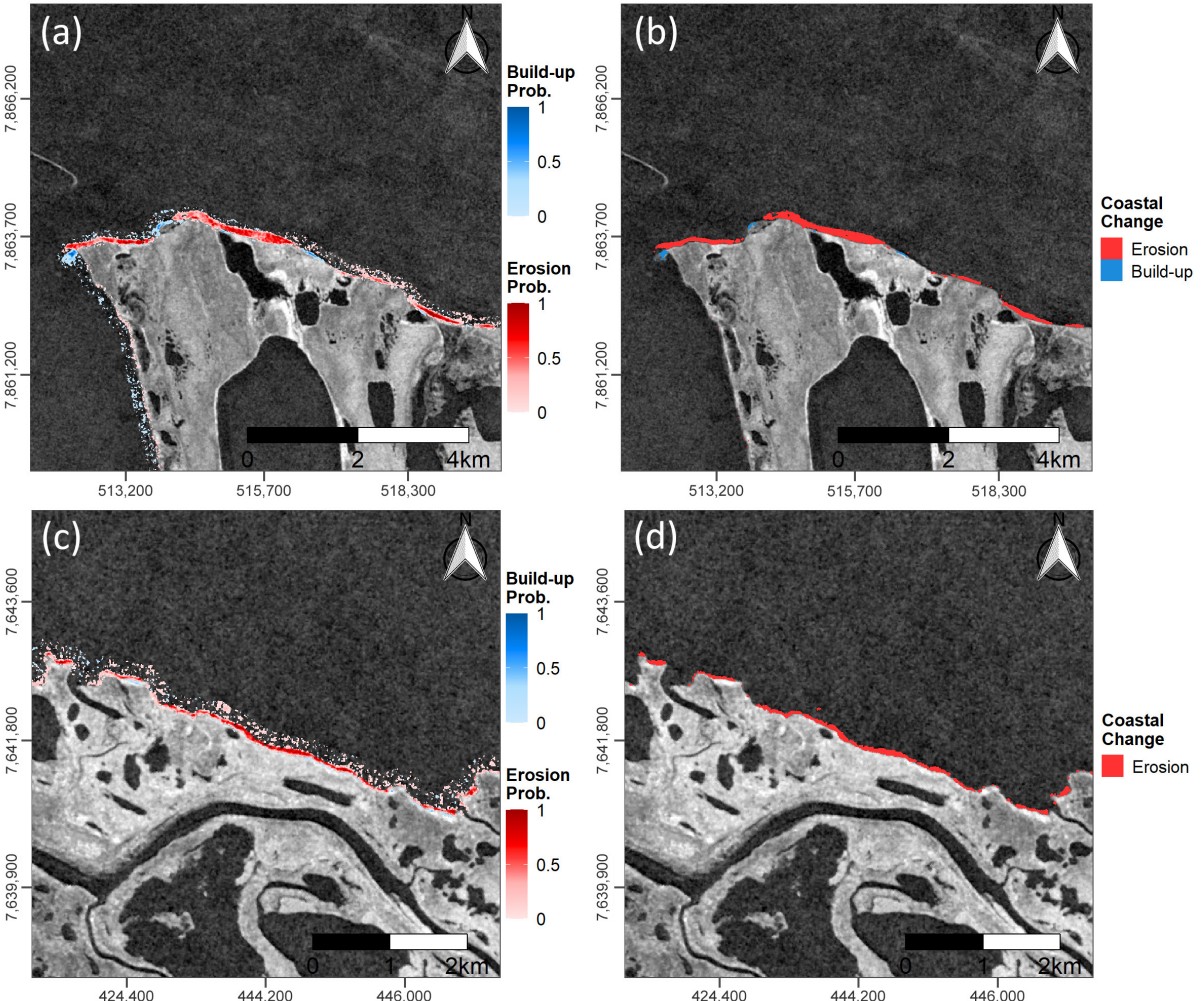

**Figure 10.** Probability maps of erosion and build-up rates between 2017 and 2020 for subsets of Cape Halkett, USA (aoi 02) (**a**), and Shoalwater Bay, Canada (aoi 03) (**c**), together with their respective maps of coastal change (**b,d**). A threshold of 0.35 for the erosion and 0.6 for the build-up was applied (**b,d**).

The CVA-based coastal change analysis revealed maximum erosion rates of up to 160.3 m and an average rate of 4.4 m across the total investigated area over three years between 2017 and 2020. Maximum build-up rates of 166.7 m and an average build-up rate of 1.9 m could be observed across all aois, respectively. The numbers hereby refer to the average rates of change for 200 m segments along the coastline. The strongest erosion was detected for Drew Point–Cape Halkett in the USA (max: 160.3 m; average: 22 m), followed by Shoalwater Bay in Canada (max: 70 m; average: 10.3 m). On the other hand, the highest build-up numbers were observed for Bezimyanniy Cape–Eastern Oyagoss Cape in Russia (max: 166.7 m; average: 8.7 m), followed by Sims Bay, Russia (max: 58 m; average 1.4 m).

A full list of the maximum and average rates of erosion and build-up per aoi is given in Table 5.

The average erosion rates between 2017 and 2020 for the 200 m segments along the coast of Shoalwater Bay, Canada, are displayed in Figure 11. The color coding of the map highlights the areas of high coastal erosion rates (reddish color) in contrast to the regions with low erosion rates (bluish colors).

**Table 5.** Average and maximum rates of erosion and build-up between 2017 and 2020 per aoi. The numbers are based on average values for 200 m segments along the coastline. Strongest erosion rates were observed for Drew Point–Cape Halkett, United States of America (USA) (aoi 02), whereas Bezimyanniy Cape–Eastern Oyagoss Cape in Russia (aoi 09) features highest build-up rates.

| AOI | Erosion Rates | | Build-Up Rates | |
|---|---|---|---|---|
| | Mean | Max | Mean | Max |
| 1 | 0.2 m | 10.3 m | 0 m | 0 m |
| 2 | 22 m | 160.3 m | 4 m | 79 m |
| 3 | 10.3 m | 70 m | 0 m | 2 m |
| 4 | 0.5 m | 7.6 m | 0.2 m | 10.2 m |
| 5 | 0.7 m | 30 m | 1.4 m | 58 m |
| 6 | 1.4 m | 62 m | 0 m | 0.5 m |
| 7 | 0.6 m | 13.6 m | 0.1 m | 14 m |
| 8 | 1.1 m | 26.2 m | 0.5 m | 7.2 m |
| 9 | 1.3 m | 20.5 m | 8.7 m | 166.7 m |
| 10 | 0.9 m | 40.5 m | 1 m | 36 m |
| All: | 4.4 m | 160.3 m | 1.9 m | 166.7 m |

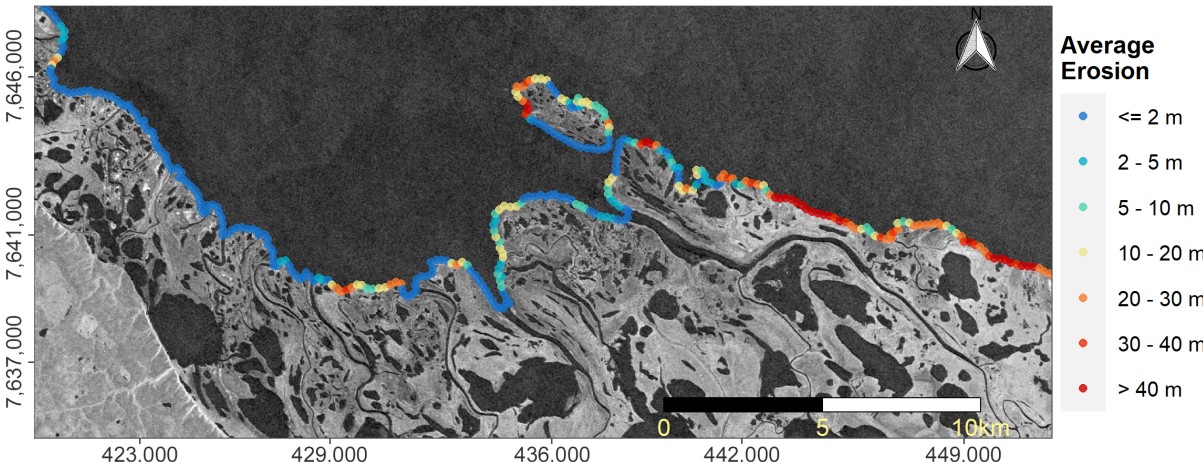

**Figure 11.** Average erosion rates between 2017 and 2020 for 200 m segments along Shoalwater Bay in Canada (aoi 03). A S1 median VV backscatter image of the year 2020 (months June–September) is used as a background image.

## 5. Discussion

The degradation of frozen ground material comes with drastic consequences not only for the environment but also for human society. Amplified surface deformation rates [71–74], wildfires [75–77], thermokarst pond and lake dynamics [78–81], coastal erosion [10,82,83], and the release of stored organic carbon content [84–87] are hereby just some of the effects associated with thawing permafrost. This study explored the potential of S1 C-Band SAR backscatter data in quantifying annual erosion and build-up rates of Arctic permafrost coasts. Ambiguities in the scattering behavior of individual SAR scenes can impair the applicability of S1 for coastal erosion analysis [26]. However, by working on

annual composites instead of individual images, the amount of noise and speckle could be reduced to a minimum. Moreover, combining images over a period of four months (June–September) minimized the geolocation uncertainty of single observations [34,35]. Because this study investigated coastal change on a per-pixel-basis with a nominal spatial resolution of 10 m, reducing the mentioned uncertainty factors is required in order to detect actual change instead of noise. A higher annual median backscatter could be observed over land compared to water, whereas the annual sd backscatter behaved inversely to the median (Figure 4). In general, water areas tend to feature lower backscatter intensities due to the specular reflection characteristic compared to the rougher terrain, which induces diffuse scattering and therefore higher backscatter values at the present incidence angles [88,89]. This is reflected in the lower median intensity over the sea in contrast to terrestrial areas. That said, the sea surface is not perfectly flat but characterized by different wave types. On the one hand, there are wind-driven capillary waves, which rely on surface tension [89]. On the other hand, gravity waves are generated by gravitational force to counteract wind-induced mass disturbances [89,90]. As each SAR observation detects a unique texture of the sea surface as a result of the present waves for a given time, a higher sd in backscatter intensity can be observed for water areas compared to the land areas, which are, for the most part, more stable in their surface roughness across the observed time span (June–September).

This information could subsequently be used as input data for generating a DL-based coastline product, which acted as a basis for the CVA-derived erosion and build-up rates. CNN-based algorithms have the drawback of requiring large amounts of training data compared to traditional ML approaches [91]. This limitation could be largely overcome by (1) using pre-trained networks based on the ImageNet database (14 Mio. images) as well as (2) applying augmentation to the additional input images which enabled further training using 32,606 pseudo-RGB SAR scenes. Another limitation is the difficulty in finding the optimal depth of a CNN for a given task [92]. In comparison to traditional machine learning approaches, which require feature engineering (the selection of relevant features as input data) for the best possible performance [93], DL identifies the relevant representation of the input data and ignores seemingly irrelevant representations by itself [94]. However, the choice of a suitable network architecture and depth capable of grasping the complexity of a problem is still essential in order to generate meaningful results [92]. By combining nine different U-Net networks with varying depths and architecture types, a representative and robust prediction per pixel is provided. As mentioned in a recent review article by Philipp et al. [95], only a relatively small portion of the reviewed studies explored the potential of DL in the context of permafrost-related investigations, despite its large potential for improving classification accuracies compared to traditional ML approaches. Several successful implementations of DL for the mapping of thaw slumps [96], ice-wedge polygons [97,98], Arctic vegetation [99], and Arctic settlements [100] were already published. This study exploited the capabilities of DL and, in particular, of the U-Net architecture for extracting the Arctic coastline with high resolution and accuracy. The successful implementation of the U-Net framework for detecting coastlines based on SAR data in Antarctic environments was hereby already demonstrated in previous works [40–42]. In this study, nine different U-Net architectures were employed to generate binary classification maps that differentiate between sea and land area (including inland lakes and rivers). All the U-Net models hereby produced validation accuracies of $\geq 0.9957$ (Table A1). Therefore, the algorithm was largely successful in differentiating between inland water bodies and sea area. By taking the mode from the nine resulting classification maps, the most representative land-cover class per pixel could be derived. Moreover, taking the mode reduced the overall amount of noise from the misclassified pixels compared to the individual classification maps of each model. Having said that, high accuracy numbers are expected for a binary classification. Because the focus of this study lies on the change of permafrost-affected coastlines, the correct identification of the boundary between the land and sea is especially of relevance. Therefore, the accuracy metrics of the final binary classification map within a

500 m buffer around the manually digitized coastline were provided. Overall, the accuracy was hereby slightly lower compared to the accuracies across the entire scene while still being high with an average value of 0.974. No significant deviations in the accuracy metrics between the training and validation areas could be observed. Moreover, the deviation of the final coastline product compared to the manually digitized reference line proved to be ±28 m on average.

Uncertainties in the classification occurred in areas where a hard differentiation between sea and land is difficult even by visual interpretation, such as flat sandy coasts. Furthermore, river deltas tend to be challenging as the algorithm might struggle to identify where the mouth of a river starts and the actual river ends. Despite these challenges, the algorithm proved to be capable even in these challenging environments, as visualized in Figure 8. Moreover, the U-Net framework proved, for the most part, to successfully differentiate between inland water bodies and the open sea area. The remaining inland lakes could be removed via a simple closing holes algorithm. Lastly, the reference data are based on satellite imagery derived from S1, S2, and Google Earth. Although in situ measurements would be favorable, working with 1038 km of manually digitized coastline based on high-resolution satellite imagery (<=10 m) as a reference is interpreted as a reasonable approach for this first step toward a circum-Arctic-scale permafrost coastal erosion analysis on a 10 m spatial resolution.

The extracted coastline featured an average accuracy of ±28 m and thus outperformed other existing, freely available, and circum-Arctic coastline datasets (Table 4) across the investigated regions. The CAVM coastline was derived from the Digital Chart of the World (DCW) dataset, released in 1992, on a scale of 1:1,000,000 [101]. Despite being one of the most comprehensive global databases of its time, it has not been updated since 1992. Moreover, the original coastline from the DCW was further simplified by, e.g., removing islands smaller than 49 km$^2$ and by combining any two lines closer than 0.5 km [102]. The GSHHG, formerly known as the Global Self-consistent, Hierarchical, High-resolution Shorelines (GSHHS), is based on the World Vector Shorelines (WVS), the CIA World Data Bank II (WDBII), and the Atlas of the Cryosphere (AC) and was last updated in the year 2017 [54,103]. OSM is a community-driven and non-commercial project that aims to have a complete record of the world's geographic features built through crowd-sourcing [104]. It is the most successful geographic information-based crowd-sourced project to date and became a popular data source [105]. Although the coastline derived from OSM featured the highest accuracy out of the three publicly available shoreline products mentioned in this study, the quality of the product varies strongly across different regions. The S1 and DL-based coastline computation process, as proposed within the framework of the study, offers not only high accuracy but also an up-to-date observation of the current state for these highly dynamic regions and can prospectively be applied for the entire Arctic.

The mostly inverse behavior of annual sd and annual median backscatter was not only useful for the DL framework but could also be exploited for the CVA. Compared to traditional post-classification change detection approaches, an accumulation of errors from the separate input classifications is avoided in the physical-based CVA approach [56]. Moreover, the CVA method tends to be more flexible and less computationally intensive in comparison to multi-date classification change detection [65]. Because the median proved to be higher over land, whereas the sd was higher over water, a change from a high median backscatter to a high sd backscatter could be interpreted as a change from land to water and, thus, as erosion. Logically, a change from a high sd to a high median could be interpreted as build-up. The resulting probability maps visualize areas of coastal erosion and build-up but also noise in the form of low probability of change pixels scattered across the water (Figure 10). Because the sea surface is not stable but features slightly different backscatter values in each observation due to the previously mentioned capillary and gravity waves [89], the annual median and sd backscatter values deviate to some degree for each year. Hence, as the CVA computes the magnitude of change in each direction [56,57], this deviation in backscatter intensity is being picked up by the probability maps. However,

the magnitude of change from land to water and vice versa is, for the most part, significantly stronger compared to the variations within the sea or terrestrial area alone. Therefore, actual coastal change can be derived by applying a threshold to the probability maps. A threshold of 0.35 for erosion and 0.6 for build-up is recommended for the investigated areas. However, optimal threshold values might change, depending on the area, coast type, and data availability. The accuracy assessment revealed that a CVA slightly underestimates the actual change with an average deviation of $-10.3$ m while at the same time favoring less amount of noise. By adjusting the thresholds, the underestimation could potentially be reduced, and small coastal change rates could be detected more securely at the risk of introducing a higher amount of noise. For this scenario, the probability maps provide a valuable reference in defining the most suitable threshold values across the abundance of different Arctic coastal environments.

The extracted erosion rates agree with numbers published in the previous literature. Especially Drew Point–Cape Halkett in Alaska featured the overall strongest average (22 m in three years; 7.3 m/year) and maximum (160.3 m in three years; 53.4 m/year) erosion rates across all the investigated areas. This matches with observations by Jones et al. [12] who investigated the annual erosion of the Drew Point coast from 2007 to 2016 via very high resolution optical satellite imagery. Annual average erosion rates ranged hereby between 6.7 and 22.6 m, with maximum annual erosion rates between 19.6 and 48.8 m [12]. Similarly, in a recent study by Wang et al. [106], erosion rates of 30.8–51.4 m/year were identified for six study locations distributed across the Drew Point coast during the time period 2009–2017 via Landsat data. In contrast, the extracted average erosion rates for other investigated regions such as Mus-Khaya Cape–Mouth of Peshanaya, Russia (aoi 06), and Bezimyanniy Cape–Eastern Oyagoss Cape, Russia (aoi 09), are comparatively small. Similar findings for these regions were reported in a study by Günther et al. [107], who employed high-resolution historical and up-to-date satellite imagery covering a time span between 1965 and 2011 to identify the average erosion rates of 2.1 m/year for Cape Mamontov Klyk (partially covered by aoi 06) and 3.4 m/year for Oyogos Yar (partially covered by aoi 09). Deviations between predicted erosion values and numbers published in the previous literature can be attributed to different spatial resolutions of applied data, non-identical observed temporal windows, as well as the size and exact locations of the investigated regions.

On the one hand, the proposed methods and data provide a valuable tool for quantifying erosion and build-up rates of Arctic permafrost coasts. On the other hand, the quality of the output product strongly depends on the amount of available S1 backscatter data, which varies over space and time [108,109]. At the time of writing this article, no data have been generated by S1B since December 23rd 2021 due to an on-board anomaly and, thus, currently having only one of the two S1 satellites active [110]. Depending on the number of available scenes, the optimal threshold might deviate from the proposed threshold values in this study. In this case, the associated probability map is a useful tool for identifying the most suitable threshold for a given area and time span. Furthermore, as mentioned by Bartsch et al. [26], SAR-specific challenges, including RADAR shadows, foreshortening, and ambiguities in the backscatter behavior, might restrict the applicability in some areas. That said, the majority of noise originating from backscatter ambiguities, geolocation uncertainties, and tidal changes was mostly averaged out by working on annual composites. Moreover, while the applied Deep Learning method generated a coastline product with high accuracy values for both the training and validation sites, more training data might be needed for a circum-Arctic application in order to account for the diversity of Arctic coastal environments. Lastly, despite working on satellite data with a relatively high spatial resolution, quantifying the rates of erosion and build-up is only meaningful if the observed change is greater than or equal to the size of one pixel, which in the case of the S1 GRD data is 10 m. However, by increasing the observation time span, the applicability of the continuously generated S1 GRD scenes also increases, even for areas with little erosion processes. Therefore, S1 is a very attractive data source for the current and future SAR-based monitoring of changes in Arctic permafrost coasts. Next to S1, the proposed methods

could also be applied to SAR data from other satellite programs, such as the RADARSAT Constellation Mission (RCM) [111]. This mission has a high potential for extracting detailed shoreline information, especially when combining the backscatter information with optical imagery from, e.g., the Landsat legacy [112].

Stronger erosion rates of Arctic permafrost coasts are reported for recent years and are expected to further increase in the future [11,12,107,113]. As mentioned in a recent review article by Irrgang et al. [11], rapid changes in Arctic coastal environments call for a coordinated and interdisciplinary effort of not only scientists but also policymakers, stakeholders, and the local population in order to develop suitable adaptation and mitigation strategies. This highlights the need for a continuous and large- to circum-Arctic-scale monitoring framework of Arctic coastlines. As demonstrated in this study, S1 imagery in combination with DL and CVA provides a powerful tool to address this challenge. The proposed methods and data can be applied on a pan-Arctic scale and the observed coastal change rates may subsequently be used as a reference for quantifying the volume loss of frozen ground, and for estimating the release of stored organic carbon content in future analyses.

## 6. Conclusions

Within this study, the potential of combining Synthetic Aperture RADAR (SAR) data derived from Sentinel-1 (S1) Ground Range Detected (GRD) scenes with a Deep Learning (DL) framework and a Change Vector Analysis (CVA) for quantifying the annual erosion and build-up rates of Arctic permafrost coasts was investigated. A total of 1038 km of Arctic coastline and an area of 19,275 km$^2$ divided into ten different regions across the Arctic was analyzed. Working on annual median and standard deviation (sd) backscatter images from 2017 to 2020 allowed for the generation of a high-quality reference coastline product via DL, and for quantifying coastal change rates via CVA. The following main conclusions can be drawn from this study:

- On average, the annual median S1 GRD backscatter was observed to be higher over terrestrial areas and lower over water. The sd behaved, for the most part, inversely to the median and featured higher values over water and lower values over land. Thus, a change from a high median backscatter to a high sd backscatter can be interpreted as a change from land to water.
- DL in combination with annual pseudo-Red Green Blue (RGB) S1 composites allowed for the computation of a high-quality reference coastline with an accuracy of $\pm$28 m across the investigated regions.
- Maximum erosion rates of up to 160.3 m between 2017 and 2020 could be observed for Drew Point–Cape Halkett in Alaska. Average erosion rates ranged from 0.2 to 22 m, depending on the study region. The average erosion across all the investigated regions proved to be 4.4 m for a three-year observation period. The observed erosion rates agree with findings published in the previous literature.
- Maximum build-up rates of up to 166.7 m were observed for Bezimyanniy Cape–Eastern Oyagoss Cape, in Russia, whereas an average build-up rate of 1.9 m across all the study regions between 2017 and 2020 can be reported.

The proposed methods and data proved to be potent tools for quantifying the annual erosion and build-up rates of Arctic permafrost coasts. The tested approach can be applied on large scales, prospectively even on a circum-Arctic scale. For circum-Arctic application, additional training of the DL framework might be necessary in order to cover the diversity of Arctic coastal environments. Furthermore, optimal thresholds for the CVA analysis might deviate, depending on the number of available satellite scenes and the coastal environment. The generated products may be of use for future analyses related to changes in Arctic coastal environments, such as quantifying the amount of lost frozen ground and the release of organic carbon content.

**Author Contributions:** M.P. and C.K. conceptualized the study design. M.P. processed, analyzed, and visualized the data and wrote the original manuscript. T.U., A.D., and C.K. contributed to the study concept, the writing, and the editing of the manuscript. All authors have read and agreed to the published version of the manuscript.

**Funding:** This publication was supported by the Open Access Publication Fund of the University of Wuerzburg.

**Institutional Review Board Statement:** Not applicable.

**Informed Consent Statement:** Not applicable.

**Data Availability Statement:** All data that support the findings of this study are available from the corresponding author upon reasonable request.

**Acknowledgments:** We would like to thank three anonymous reviewers and the academic editor for their helpful comments on this manuscript.

**Conflicts of Interest:** The authors declare no conflict of interest. The funders had no role in the design of the study; in the collection, analyses, or interpretation of data; in the writing of the manuscript; or in the decision to publish the results.

## Abbreviations

The following abbreviations are used in this manuscript:

| | |
|---|---|
| AC | Atlas of the Cryosphere |
| ACD | Arctic Coastal Dynamics |
| aoi | area of interest |
| BN | Batch Normalization |
| CAVM | Circumpolar Arctic Vegetation Map |
| CNN | Convolutional Neural Network |
| Conv | Convolution |
| CVA | Change Vector Analysis |
| dB | decibel |
| DCW | Digital Chart of the World |
| DEM | Digital Elevation Model |
| DL | Deep Learning |
| GEE | Google Earth Engine |
| GRD | Ground Range Detected |
| GSHHG | Global Self-consistent, Hierarchical, High-resolution Geography Database |
| GSHHS | Global Self-consistent, Hierarchical, High-resolution Shorelines |
| IW | Interferometric Wide |
| ML | Machine Learning |
| MV | Moving Window |
| OSM | OpenStreetMap |
| RCM | RADARSAT Constellation Mission |
| ReLU | Rectified Linear Unit |
| RGB | Red Green Blue |
| RMSprop | Root Mean Square Propagation |
| S1 | Sentinel-1 |
| S2 | Sentinel-2 |
| SAR | Synthetic Aperture RADAR |
| sd | standard deviation |
| USA | United States of America |
| WDBII | CIA World Data Bank II |
| WVS | World Vector Shorelines |

## Appendix A

**Table A1.** Accuracy statistics and epochs of final segmentation maps per model. The epoch with the highest validation accuracy was chosen as a representation for each model. Accuracy and loss values were rounded to the fourth decimal place.

| Model | Epoch | Training Acc. | Training Loss | Validation Acc. | Validation Loss |
|---|---|---|---|---|---|
| VGG16 | 22 | 0.9991 | 0.003 | 0.9979 | 0.0085 |
| VGG19 | 24 | 0.9993 | 0.0027 | 0.998 | 0.0075 |
| ResNet34 | 29 | 0.9991 | 0.003 | 0.998 | 0.0087 |
| ResNet50 | 14 | 0.9977 | 0.0082 | 0.9969 | 0.0116 |
| Inception v3 | 29 | 0.9997 | 0.001 | 0.9957 | 0.0167 |
| Inception-ResNet v2 | 26 | 0.9992 | 0.0025 | 0.9979 | 0.0081 |
| ResNeXt | 20 | 0.9986 | 0.0053 | 0.9967 | 0.0113 |
| DenseNet121 | 15 | 0.9977 | 0.0091 | 0.9976 | 0.0111 |
| SE-ResNeXt50 | 25 | 0.9998 | 0.0005 | 0.9964 | 0.0117 |

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
