# Peer review of "Automated Extraction of Annual Erosion Rates for Arctic Permafrost Coasts Using Sentinel-1, Deep Learning, and Change Vector Analysis"

_remotesensing, doi:10.3390/rs14153656_

Round 1

Reviewer 1 Report

This is a very interesting article about changes in the coastline in areas of persistent permaphrost.

However, I have a few comments, indicated below:

Line 82 - Study Area, please explain why only 10 areas were selected for analysis. Do they represent all the polar regions? There are no western regions - Norway and Denmark. Moreover - why the authors did not indicate at least 1 place in the Southern Hemisphere?

Figure 4, 6, 7, 8, 9. Interesting, however, the cartographic projection does not reflect the region that is presented, a better projection would be WGS84. Providing a region name (aoi) would be preferable to an area number.

Table 2. Why did the authors use only 2 measurement periods, over a of 3 years period? Was the coastline unchanged in the remaining years?

Table 6. Interesting results, but no verification of the correctness of the calculations. Some areas do not have a descending path. The authors did not take into account other missions, did not compare the results to other period 

Interesting results, but no verification of the correctness of the calculations. Some areas do not have a descending path. The authors did not take into account other missions, did not compare the results to other missions. A scientific study requires the control of the results.

Interesting discussion, only whether it is possible to assess the annual erosion of the polar coasts from 2 time points. Can it be inferred from 10 places about the state of the polar coast?

Greetings,

Author Response

Dear reviewer, thank you for taking the time to review our manuscript. Thanks to the helpful comments we were able to further improve the quality of our manuscript. Please find attached a PDF file with our replies to each of the reviewers' comment.

Reviewer 2 Report

Comments on remotesensing-1656582

This study lacks scientific/engineering factors. It is more like a report of datasets uploaded to a global data platform. The two methods applied are empirical and controversial, actually, and the authors fail to demonstrate any one piece of in-situ manually measured evidence to persuade readers. The claimed resolution of satellite data and comparison with reference lines are circumstantial. That is why they observed the predicted coastline runs closely to the reference line. Data calibrations are vital in any scientific/engineering measurement and field survey. I suggest the authors feature their research theme more academically. Below are a few opinions for the authors to consider.

  1. A general introduction of the deep learning (DL) model, specifically convolutional neural networks, should be given and discussed. Simple mathematical derivations of the algorithm may help. By the same token, a brief discussion of at least one of the other parallel algorithms of Change Vector Analysis (CVA) is required to clarify why the combination of DL and CVA is selected in this study.
  2. The tasks of the deep learning algorithm in this study need to be clearly addressed. Are they supervised or unsupervised?
  3. The theories of DL and CVA are incomplete. The authors are supposed to explain this issue. I believe that the authors’ “products” are empirical.
  4. DL architectures always display challenging performances due to limitations in their internal representations. The authors seem to ignore this theoretical difficulty and impute it to data complexity.
  5. Performing the CVA for erosion investigation is easier than other algorithms in the beginning stage, however, the challenge is the selection of appropriate change thresholds and interpretation. Therefore, the processing cost is about the same as another comparable method, direct multi-date classification (DMC). Is there any special reason that the authors prefer the CVA rather than DMC?
  6. A comparison with any parallel method is essential when introducing a new method or processing scheme. The authors are required to elaborate on this issue. A simple example of real data or simulation would help.

Author Response

(The authors gave the same response as above.)

Reviewer 3 Report

This manuscript presents an interesting study on automated extraction of annual erosion rates for Arctic permafrost coast based on Sentinel-1 data sets and the approaches including deep learning and change vector analysis. In general, the study is well-structured and –written, and the results are reasonably presented and discussed. I suggest the manuscript can be accepted for publication after a minor revision.

Some minor issues:

Line 14, annual rate should be presented with a unit “per year”, otherwise it just represents the changes between two years.

Line 30, “said”  is this a typo?

Line 99, architecures, typo

Line 100, Said, typo?

Author Response

(The authors gave the same response as above.)

Round 2

Reviewer 2 Report

Comments on remotesensing-1656582 V2

The manuscript has improved some, but basic problems, before and after revision, must be solved before considering publication.

  1. The authors must try their best to learn decent knowledge of employing abbreviations in technical writing. Pay attention to the different conventions of applying abbreviations/acronyms in the abstract and main body, the format consistency, and the timing of appearance. Refer to a good writer’s manual for more details.
  2. Equation (5) is questionable and may be incorrect mathematically. Cite the origin if it is copied from somewhere. I strongly suggest the authors perform a little manual calculation to verify the equation because the original documents may be wrong.
  3. We believe that any algorithm has advantages and shortcomings, no matter how robust it is. Most readers trust the contents of the published material, therefore, it is the responsibility of authors to present the truth as much as possible because the authors who utilize the algorithm know the facts. This suggestion applies particularly to the users of any empirical-based approach. Even though it has been successfully applied to numerous cases in various fields, we never know what would happen in the next case. Only with a sound theoretical foundation and rigorous mathematical verification can we unify the diverse observations and make an objective evaluation of the validity of the empirical results, and deduce principles or laws thereafter. Although the authors have provided a long argument pertaining to this issue, it is more like to defend their approach by imputing the difficulties to the shortage of data and the complexity of the case instead of examining the intrinsic incapability of the approach. My experiences taught me that “good data don’t need good methods”.

In short, I will be happy to see the authors discussing their true  understanding of the shortcomings of their proposed method, and possible remedies. It is wise to address that the success of the proposed method, if there is any, is limited to the case investigated.  

Author Response

We want to thank the reviewer for taking the time to review our manuscript. Please find our detailed answers to each comment/suggestion in the attached file.
